# ON DISENTANGLED REPRESENTATIONS LEARNED FROM CORRELATED DATA

## ABSTRACT

Despite impressive progress in the last decade, it still remains an open challenge to build models that generalize well across multiple tasks and datasets. One path to achieve this is to learn meaningful and compact representations, in which different semantic aspects of data are structurally disentangled. The focus of disentanglement approaches has been on separating independent factors of variation despite the fact that real-world observations are often not structured into meaningful independent causal variables. In this work, we bridge the gap to real-world scenarios by analyzing the behavior of most prominent methods and disentanglement scores on correlated data in a large scale empirical study (including 4260 models). We show that systematically induced correlations in the dataset are being learned and reflected in the latent representations, while widely used disentanglement scores fall short of capturing these latent correlations. Finally, we demonstrate how to disentangle these latent correlations using weak supervision, even if we constrain this supervision to be causally plausible. Our results thus support the argument to learn independent mechanisms rather than independent factors of variations.

## 1 INTRODUCTION

Due to the induced structure, disentangled representations promise generalization to unseen scenarios (Higgins et al., 2017b), increased interpretability (Adel et al., 2018; Higgins et al., 2018) and faster learning on downstream tasks (van Steenkiste et al., 2019; Locatello et al., 2019a). While the advantages of disentangled representations have been well established, they generally assume the existence of natural factors that vary independently within the given dataset, which is rarely the case in real-world settings. As an example, consider a scene with a table and some chairs (see Fig. 1). The higher-level factors of this representation are in fact correlated and what we actually want to infer are independent (causal) mechanisms (Peters et al., 2017; Parascandolo et al., 2018; Suter et al., 2019; Goyal et al., 2019).

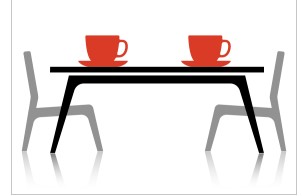

Figure 1: While in principle we consider the presence of the objects (coffee cup, table, chair) to be independent mechanisms, they tend to appear together in observed data.

A complex generative model can be thought of as the composition of independent mechanisms or "causal" modules, which generate high-dimensional observations (such as images or videos). In the causality community, this is often considered a prerequisite to achieve representations which are robust to interventions upon variables determined by such models (Peters et al., 2017). One particular instantiation of this idea in the machine learning community is the notion of *disentangled* representations (Bengio et al., 2013). The goal of disentanglement learning is to find a representation of the data which captures all the ground-truth factors of variation (FoV) independently.

Despite the recent growth of the field, the performance of state-of-the-art disentanglement learners remains unknown for more realistic settings where FoV are correlated during training. Given the potential societal impact in the medical domain (Chartsias et al., 2018) or fair decision making (Locatello et al., 2019a; Madras et al., 2018; Creager et al., 2019), the evaluation of the usefulness of disentangled representations trained on correlated data is of high importance.

To go beyond the highly idealized settings considered thus far, we conducted a large scale empirical study to systematically assess the effect of induced correlations between pairs of factors of variation in training data on the learned representations. To provide a qualitative and quantitative evaluation, we investigate multiple datasets with access to ground-truth labels. Moreover, we study the generalization abilities of the representations learned on correlated data as well as their performance in particular for the downstream task of fair decision making.

**Contributions.** Our main contributions can be summarized as follows:

- We present the first large-scale empirical study (4260 models)[1] that examines how modern disentanglement learners perform when ground truth factors of the observational data are *correlated*.
- We find that factorization-based inductive biases are insufficient to learn disentangled representations from observational data. Existing methods fail to disentangle correlated factors of variation, resulting in correlated latent space dimensions. Moreover, standard disentanglement metrics do not reveal these persisting correlations.
- We investigate the usefulness of semi-supervised and weakly-supervised approaches to resolve latent entanglement. For the latter setting, we focus on multiple observational and interventional distributions.

## 2 BACKGROUND AND RELATED WORK

**Disentanglement.** Current state-of-the-art disentanglement approaches use the framework of variational auto-encoders (VAEs) (Kingma & Welling, 2014; Rezende et al., 2014). The (high-dimensional) observations $x$ are modelled as being generated from some latent features $z$ with chosen prior $p(z)$ according to the probabilistic model $p_\theta(x|z)p(z)$. The generative model $p_\theta(x|z)$ as well as the proxy posterior $q_\phi(z|x)$ can be parameterized by neural networks, which are optimized by maximizing the variational lower bound (ELBO) of $\log p(x_1, \ldots, x_N)$.

$$\mathcal{L}_{VAE} = \sum_{i=1}^{N} \mathbb{E}_{q_\phi(z|x^{(i)})}[\log p_\theta(x^{(i)}|z)] - D_{KL}(q_\phi(z|x^{(i)})\|p(z)) \qquad (1)$$

The above objective does not enforce any structure on the latent space, except for similarity (in KL-divergence) to the prior $p(z)$ (typically chosen as an isotropic Gaussian). However, the structure and semantic meaning of latent representations can be relevant to study generation properties. Consequently, various proposals for structure-imposing regularization and commonly used evaluation metrics measuring different notions of disentanglement of the learned representations have been made (Higgins et al., 2017a; Kim & Mnih, 2018; Burgess et al., 2018; Kumar et al., 2018; Chen et al., 2018; Eastwood & Williams, 2018; Mathieu et al., 2018). Recently, it has been shown that unsupervised disentangling by optimising marginal likelihood in a generative model is impossible without further inductive bias (Locatello et al., 2019b). To address this theoretical limitation, methods have been proposed that do not require explicitly labelled data but only some weak labeling information (Locatello et al., 2020; Shu et al., 2019). Ideas related to disentangling the factors of variations date back to the non-linear ICA literature (Bach & Jordan, 2002; Comon, 1994; Jutten & Karhunen, 2003; Hyvärinen & Pajunen, 1999; Hyvarinen et al., 2019; Hyvarinen & Morioka, 2016; Gresele et al., 2019). Recent work combines non-linear ICA with disentanglement (Khemakhem et al., 2020; Sorrenson et al., 2020; Klindt et al., 2020)

**Correlations.** A set of random variables $X_{i=1,\ldots,n}$ is not independent, if and only if their joint distribution does not factorize

$$P(X_1, X_2, \ldots, X_n) \neq \prod_{i=1}^{n} P(X_i). \qquad (2)$$

In this case, we speak of dependence between the random variables, also commonly referred to as correlation.[2] Correlation between two variables can either stem from a direct causal relationship (one

---

[1]Each model has been trained for 300,000 iterations on Tesla V100 GPUs. Reproducing these experiments requires approximately 0.79 GPU years.

[2]We use the term correlation here in a broad sense of any statistical association, not just linear dependencies.

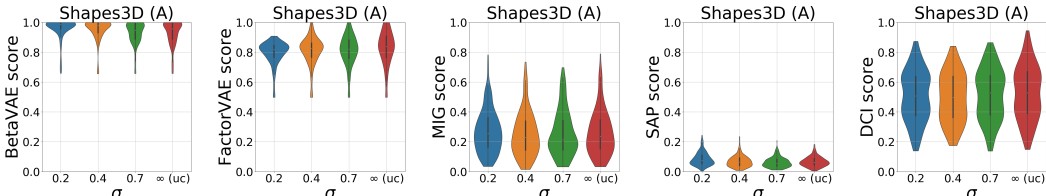

Figure 2: Disentanglement metrics show no clear trend along different correlation strengths (each violin represents 180 models, higher $\sigma$ indicates less correlation).

variable affects the other), but can also be due to different unobserved circumstances (confounders) affecting both. Real-world datasets display many of these ("spurious" and often a priori unknown) correlations (Geirhos et al., 2020). However, most work on learning disentangled representations assumes that there is an underlying set of independent ground truth variables that govern the generative process of observable data. These methods are hence predominantly evaluated on data that obey independence in the true factors of variation, which we then consider to be the correct factorization. In the real world, the observation generating process is likely not always as clearly "disentangled" and we do expect correlations in the collected datasets. It is thus an open question to what degree existing inductive biases from the encoder/decoder architecture, but more importantly the dataset biases, affect the learned representation. In our experiments, we introduce dataset correlations in a controlled manner to understand to what degree state-of-the-art approaches can cope with such correlations. We believe these correlations to reflect a major feature of more realistic environments.

**Other Related Work.** Most popular datasets in the disentanglement literature exhibit perfect independence in their FoV. At some level this is sensible as it reflects the underlying assumption in the inductive bias being studied. However, this assumption is unlikely to hold in practice as shown by Li et al. (2019), who propose methods based on a pairwise independence assumption instead. The literature so far has not thoroughly measured how popular inductive biases such as factorized priors behave when learning from correlated datasets, although several smaller experiments along these lines can be acknowledged. Chen et al. (2018) studied correlated 3DFaces (Paysan et al., 2009) by fixing all except three factors in which the authors conclude that the $\beta$-TC-VAE regularizer can help to disentangle imposed correlations. Brekelmans et al. (2019) show that Echo noise results in superior disentanglement compared to standard betaVAE in a small experiment on a downsampled dSprites variant where randomly selected factor pairs are excluded. However, the latent structure was not studied in detail; our findings suggest that global disentanglement metrics are insufficient to diagnose issues when models learn from correlated data. Creager et al. (2019) based some of the evaluations of a proposed new autoencoder architecture in the fairness context on a biased dSprites variant and Yang et al. (2020) study a linear SCM in a VAE architecure on datasets with dependent variables. However, their studies focused on representation learners that require strong supervision via FoV labels at train time.

## 3 THE EFFECT OF CORRELATED DATA ON DISENTANGLEMENT LEARNERS

In this section, we want to present the key findings from our empirical study of unsupervised disentanglement learning on a particular variant of correlated data. We start by outlining the experimental design of our study in Section 3.1. Based on this, we analyze the latent spaces in Section 3.2 and find that factorization-based inductive biases are insufficient to learn disentangled representations from observational data. Persisting pairwise correlations in the latent space are not sufficiently revealed by standard disentanglement metrics that might be particularly relevant and problematic for fairness applications. Finally, in Section 3.3, we show extrapolation and generalization capabilities of the learned representations towards unseen factor combinations due to the induced correlations.

### 3.1 EXPERIMENTAL DESIGN

For our first experiments we introduce correlations between single pairs of factors of variation on the following three datasets: Shapes3D with object size and azimuth (denoted "A"), dSprites with orientation and X-position ("B") and finally the real-world observations dataset MPI-3D with

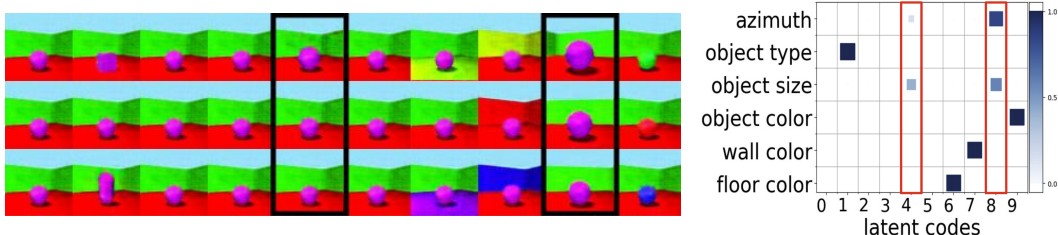

Figure 3: We show latent traversals (left) of the best DCI score model among all 180 trained models with strongest correlation ($\sigma = 0.2$) on Shapes3D (A). The traversals in latent code dimensions 4 and 8 (highlighted in black), suggest that these dimensions encode a mixture of azimuth and object size, reflecting one major axis along the correlation line of the joint distribution and one smaller, locally orthogonal axis. This is supported by a heat map of the GBT feature importance matrix of this model (right) indicating an entanglement of azimuth and object size encoded into both latent codes.

first and second degree of freedom ("C"). We focus on linear correlations with Gaussian noise between the two variables, which we denote by $c1, c2$. The ground truth factors for $c1, c2$ take values $z_{c1} \in \{0, \dots, z_{c1}^{\max}\}$ and $z_{c2} \in \{0, \dots, z_{c2}^{\max}\}$ respectively. We then parameterize correlations by sampling the training dataset from the joint $P(z_{c1}, z_{c2}) \propto \mathcal{N}(z_{c2} - \alpha z_{c1}, \sigma)$ where $\alpha = z_{c2}^{max}/z_{c1}^{max}$. The strength of the correlations can be tuned by $\sigma$, for which we choose 0.2, 0.4, 0.7 in normalized units with respect to the range of values in $z_{c1,c2}$. Lower $\sigma$ indicates stronger correlation. See Fig. 5 for an example of $P(z_{c1}, z_{c2})$ for correlating azimuth and object size in Shapes3D with $\sigma = 0.2$. Additionally, we study the uncorrelated limit ($\sigma = \infty$), which amounts to the case typically studied in the literature. We train the same six VAE methods as discussed in Locatello et al. (2019b), including $\beta$-VAE, FactorVAE, AnnealedVAE, DIP-VAE-I, DIP-VAE-II and $\beta$-TC-VAE, each with six hyperparameter settings. Each method has been trained using five different random seeds. All remaining factors of variation are sampled uniformly at random. This first study sums up to a total of 2160 trained models, or 180 models per dataset and correlation strength[3]. Appendix A describes additional implementation details.

## 3.2 CAN UNSUPERVISED METHODS ACHIEVE DISENTANGLEMENT OF CORRELATED DATA?

**Shortcomings of existing metrics.** Following recent studies, we evaluate the trained models with the help of a broad range of disentanglement metrics that aim at quantifying overall success by a single scalar measure. Perhaps surprisingly, Fig. 2 shows no clear trend among all implemented disentanglement scores w.r.t. correlation strength (see Fig. 9 in the Appendix for the full result across all datasets and metrics). The metrics have been evaluated by sampling from the correlated data distribution although they do not differ substantially when evaluated on the uncorrelated distribution. However, as we will demonstrate along the following analysis, the latent spaces in this setting show some characteristic differences when trained on a strongly correlated pair of FoVs. We thus argue that common disentanglement metrics are limited when correlations are introduced into the training data. To conduct a more careful analysis of the inductive data bias applied on the learned representations we will instead evaluate pairwise metrics. Note that regarding BetaVAE and FactorVAE this observed trend is to some degree expected as they would yield perfect disentanglement scores even if we would take the correlated ground truth factors or a linear transformation in the case of BetaVAE as the representation.

**Latent structure and pairwise entanglement.** We start by analysing latent traversals of some trained models on Shapes3D (A). For strong correlations ($\sigma = 0.2$ and $\sigma = 0.4$), we typically observe trained models with two latent codes encoding the two correlated variables simultaneously. In these cases, one of the latent codes corresponds to data along the major axis of the correlation line whereas the other latent code dimension manifests in an orthogonal change of the two variables along the minor axis. Still, a full traversal of the code corresponding to the minor axis often seems to cover only observations within the correlation line. Fig. 3 (left) shows this effect for the latent space of a model trained on Shapes3D (A) with strongest correlation ($\sigma = 0.2$).

---

[3]Code for all experiments will be released after publication

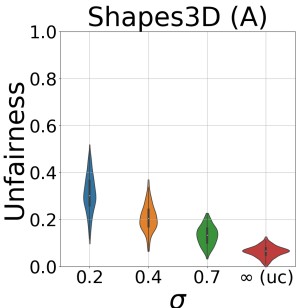

| Correlation strength | | $\sigma = 0.2$ | $\sigma = 0.4$ | $\sigma = 0.7$ | $\sigma = \infty$ (uc) |
|---|---|---|---|---|---|
| Shapes3d (A) | object size - azimuth | 0.38 | 0.26 | 0.13 | 0.08 |
| | median uncorrelated pairs | 0.09 | 0.09 | 0.09 | 0.08 |
| dSprites (B) | orientation - position x | 0.17 | 0.16 | 0.14 | 0.11 |
| | median uncorrelated pairs | 0.13 | 0.13 | 0.13 | 0.13 |
| MPI3D (C) | First DOF - Second DOF | 0.2 | 0.19 | 0.17 | 0.16 |
| | median uncorrelated pairs | 0.16 | 0.16 | 0.15 | 0.15 |

Figure 4: Left: Pairwise entanglement scores help to uncover still existent correlations in the latent representation. Left: Mean of the pairwise entanglement scores for the correlated pair (red) and the median of the uncorrelated pairs. We see that stronger correlation leads to statistically more entangled latents compared to the baseline score without correlation (blue). Right: The same behavior can be seen for the unfairness score between the correlated pair of factors.

To quantify this observation, we analyze the importance of individual latent codes in predicting the value of a given ground truth FoV. An importance weight for each pair of {FoV, latent dimension} is computed by training a gradient boosting trees (GBT) classifier to predict the ground truth labels from the latents (10,000 samples). In the right panel of Fig. 3, we compute these importance weights for the model used to generate traversals in the left panel. The corresponding evaluation for a model trained on the same dataset with weak correlation does not reveal this feature visually (see Fig. 10 in the Appendix).

To support this claim for a larger set of models, we calculate a pairwise entanglement score that allows us to measure how difficult it is to separate two factors of variation from their latent codes. This computation involves grouping FoV into pairs based on an ordering of their pairwise mutual information or GBT feature importance between latents and FoV; we defer to Appendix A for a detailed description of this procedure. Figure 4 (left) shows that across all datasets the pair of correlated FoV has a higher score than the median of all other pairs, indicating that they are harder to disentangle. This threshold decreases with weaker correlation and the pair becomes easier to disentangle for weaker correlations ($\sigma \geq 0.7$). These findings suggest that the models still manage to disentangle correlated factors if the correlation is not too strong.

Finally, correlations between variables are of crucial importance in fairness applications motivating an additional investigation on ramifications of these entangled latent spaces. In this setting we are interested in the unfairness of predicting the second correlated variable while the first correlated variable is considered being a protected or sensitive attribute. In the following, we use a variant of demographic parity (Dwork et al., 2012) that computes pairwise mutual information between latents and FoV (Locatello et al., 2019a). In Fig. 4 (right) we evaluate this score when correlations are present within the data in the case of Shapes3D (A). Unfairness tracks correlation strength in this scenario. These results suggest that we cannot expect disentangled representations learned unsupervisedly to help reduce unfairness beyond the benefits discussed in Locatello et al. (2019a). More comprehensive results, that support the finding that the correlated pair is statistically more entangled in the latent representations across all unsupervised experiments and datasets is provided in Appendix B.1.

**Summary.** We find that existing methods fail to learn disentangled representations of correlated factors of variation, and moreover that standard disentanglement metrics are insufficient to reveal these troublesome pairwise entanglements in the latent space.

### 3.3 GENERALIZATION PROPERTIES

In this Section, we aim to understand how the trained models perform on unseen training data far away from the correlation line; out-of-distribution (OOD) w.r.t. the train distribution. We analyse this capability for the model from Fig. 3. In this model, the remaining factors seem to be disentangled well enough to only focus on the two latent dimensions encoding the entangled variables.

As a first test, we sample observations from the FoV then set object size and azimuth to six distinct configurations of zero probability, see Fig. 5 (left). The trained model is capable of reconstructing these observations despite never having encountered these configurations or neighbors thereof during

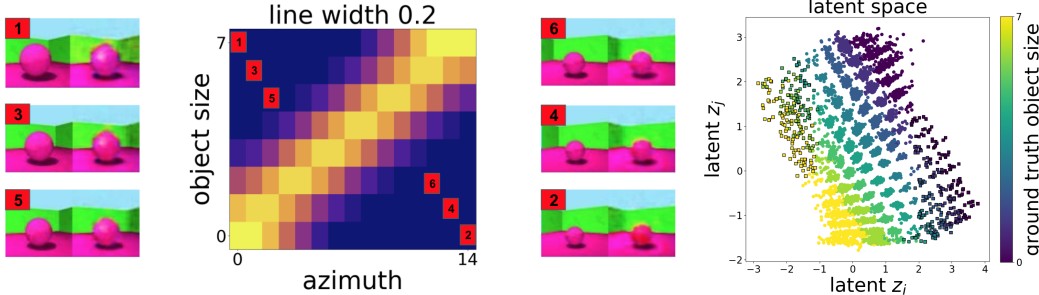

Figure 5: Generalization to out-of-distribution (OOD) test data. Left: Reconstructions of observations the model has never seen during training. Right: Latent space distribution of the two entangled dimensions. Circles without edges indicates encoded data from the (correlated) training distribution, while circles with edges indicate encoded OOD data where the correlation pattern is broken.

training. This suggests that the encoder maps representations to a meaningful point in the latent space from which the decoder is equally capable of generating expected observations. To test this hypothesis further, we analysed latent traversals originating from these OOD points and observe that changes in the remaining factors reliably yield the expected reconstructions. Traversals with respect to the two entangled latent codes continue to encode object size and azimuth.

To fully understand this models' generalization properties we visualise the occupied latent space spanned by the two identified dimensions encoding both correlated factors in Fig. 5 (right). We are particularly interested in where these points are located with respect to the ground truth value of each correlated variable, depicted via color. The two sets of depicted points are (1) latent codes sampled from the correlated training data and (2) latent codes sampled with a (object size, azimuth) configuration that has zero probability under the correlated training distribution. We observe that contours of equal color (ground truth) are not aligned with the latent axes. This indicates that the two latent dimensions encode both FoV at the same time. Likewise, we can understand the generalization capabilities of this model far away from the trained data. Extreme configurations such as small azimuth and large object size are encoded to space regions corresponding to the intersections of the manifolds with constant value for each correlated variable. This shows that all out-of-distribution points are encoded in this representation space in a way that obeys the natural ordering of each respective factor. This behaviour remains even in cases where the trained latent space does not mirror the default value ordering as stored in our ground truth table. For this we additionally trained 360 models on two additional Shapes3D variants each, where we strongly correlated object color - object size ("D") and object color - azimuth ("E") respectively ($\sigma = 0.2$ and $\sigma = 0.4$). As the color values do not allow for a unique natural ordering, the trained models do often encode a different color manifold ordering into the latent space. In Appendix B.1, we show some of their characteristic latent space visualizations with similar extrapolation and generalization capabilities. We conclude from these results that disentanglement methods can generalize towards unseen FoV configurations as long as each factor value is contained in the training data within a different configuration.

## 4 FINDING THE RIGHT FACTORIZATION

The results from Section 3 illustrate the limitations of state-of-the-art unsupervised methods on correlated data (and thus real-world observational data). We now investigate the usefulness of several approaches for mitigating pairwise correlations in the latent code. We begin with a post-hoc procedure in Section 4.1 that uses limited label information on the ground truth factors and show that it achieves a substantial correction of the pairwise latent correlation. We then consider a recently proposed approach leveraging recent advances in weakly supervised disentanglement learning that applies directly at train time. As will be seen in Section 4.2, this method results in substantially more disentangled representations, even when applied on correlated data from different sampling scenarios.

| # Labels | | 0 | 100 | 1000 | 10000 |
|---|---|---|---|---|---|
| Shapes3D (A) $\sigma = 0.2$ | object size - azimuth | 0.38 | 0.17 | 0.15 | 0.15 |
| Shapes3D (A) $\sigma = 0.2$ | median uncorrelated pairs | 0.09 | 0.08 | 0.07 | 0.07 |
| Shapes3D (A) $\sigma = 0.4$ | object size - azimuth | 0.26 | 0.1 | 0.1 | 0.1 |
| Shapes3D (A) $\sigma = 0.4$ | median uncorrelated pairs | 0.09 | 0.08 | 0.08 | 0.08 |

Figure 6: **Fast adaption with few labels:** Pairwise entanglement scores for correlated FoV pair in Shapes3D (A). The correlated pair is highlighted (red). Zero labels reflects the unsupervised baseline without any fast adaptation. Growing number of labels show that fast adaption using linear regression reduces these correlations with as little as 100 labels (blue). Reported pairwise scores are averaged over 180 models per correlation strength.

## 4.1 Post-hoc alignment correction with few labels

When a limited number of FoV labels $\{\mathbf{y}_i\}$ can be accessed, a reasonable option for resolving entangled dimensions of the latent code is by *fast adaptation*. To identify the two entangled dimensions $(z_i, z_j)$ we look at the maximum feature importance for a given FoV from a GBT trained using these labels only. We then train a *substitution function* using supervised learning to replace these two dimensions with the predicted ground truth label. Crucially, both steps of this procedure rely on the same FoV labels, which should be as few as possible. In Fig. 6 we show the pairwise entanglement score of the correlated FoVs under this fast adaptation with a linear regression as the substitution function, which succeeds with as few as 100 labels, corresponding to less than 0.02% of all data points in Shapes3D. However, fast adaptation with linear regression substitution fails in some settings: when no two latent dimensions encode the applied correlation isolated from the other latent codes, or when the correlated variables do not have a unique natural ordering (e.g. color or categorical variables). To address this, a nonlinear substitution function such as a MLP can reduce this pairwise entanglement to a certain degree (see additional results in Appendix B.2).

We find that the efficacy of fast adaptation depends on the level of disentanglement of the representations with respect to all the other factors. This implies that if the representation is well disentangled at the start of the fast adaption procedure, it is possible to achieve a perfectly disentangled model (according to our previous visual and quantitative evaluations). However, if all FoV are entangled at the beginning, the fast adaption method will have little effect. Finally, we note that model selection is impossible in a purely unsupervised manner based on any of the used disentanglement metrics, as they all require labeled ground truth data. These shortcomings shall be resolved by the following method capable of disentangling the correlated factors of variation much more reliably.

## 4.2 Alignment during training using weak supervision

Since the unsupervised disentangling by optimising marginal likelihood in a generative model is impossible (Locatello et al., 2019b, Theorem 1), inductive biases like grouping information (Bouchacourt et al., 2018) or access to labels (Locatello et al., 2019c) is required. Changes in natural environments, which typically correspond to changes of only a few underlying factors of variation, provide a weak supervision signal for representation learning algorithms (Goyal et al., 2019; Földiák, 1991; Schmidt et al., 2007; Bengio et al., 2019). Without correlations it has been shown that this weak supervision helps in learning much more disentangled representations (Locatello et al., 2020; Shu et al., 2019). Locatello et al. (2020) showed access to observations which display differences in a known number of factors of variation (without knowing which ones specifically) is sufficient to learn disentangled representation. These additional weak assumptions render the generative model identifiable in contrast to unsupervised disentanglement. This kind of extra knowledge might be available in certain settings [4], e.g., in temporarily close frames from a video of a moving robot arm where some factors remain unchanged. Hence, we want to investigate the usefulness of such a weakly-supervised method applied in various scenarios when training data is correlated. Specifically we implement the Ada-GVAE variant of Locatello et al. (2020) that was shown to allow for model

---

[4]On the other hand, in applications with fairness concerns it may be impossible to intervene on FoV representing sensitive and immutable attributes of individuals (gender, race, etc.); we refer to Madras et al. (2019) for a more complete discussion.

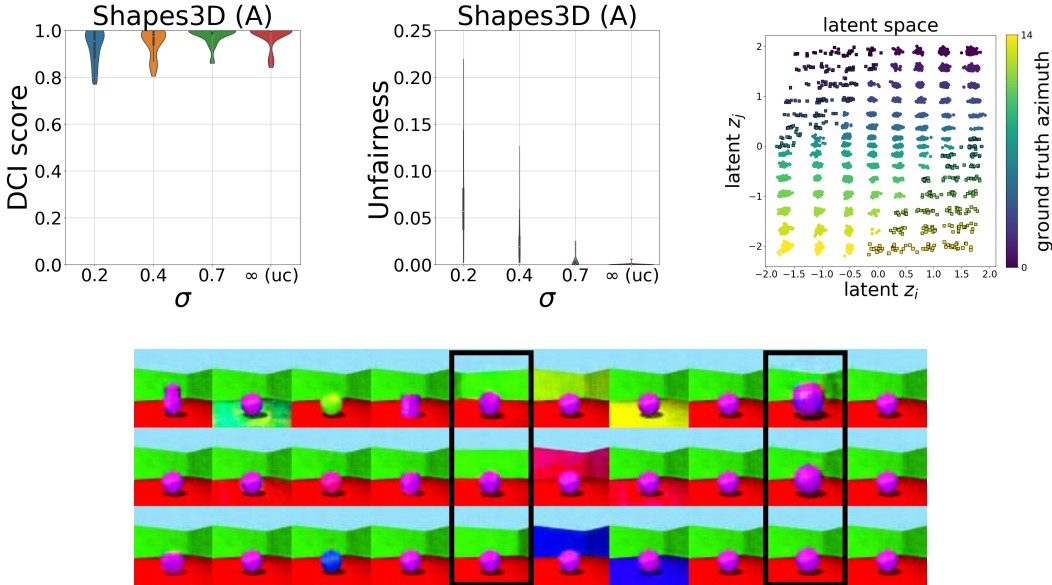

Figure 7: **Weak supervision:** Left: With weak supervision trained models on Shapes3D correlation object size and azimuth learn consistently improved, often perfect, disentangled representations across all correlation strengths. Middle: Unfairness scores between correlated FoVs are much smaller (see scale). Right: Latent dimensions of a best DCI model with strong correlation ($\sigma = 0.2$). Representations are axis-aligned with respect to both of the correlated variables ground truth values (right). Below: Traversals of a model trained with strong correlation ($\sigma = 0.2$)

selection via the (unsupervised) reconstruction loss. The method requires a pair of observations that differs in a known number of factors, without knowing which in particular.

**Weak supervision mitigates pairwise latent entanglement.** We trained the three correlated Shapes3D variants (A, D, E) with pairwise correlations between object color, object size and azimuth with the same correlation strength settings. Due to the definition of the regularizer we limit this study to the $\beta$-VAE models with the same 6 hyperparameters and use 5 random seeds, yielding 360 additional models. For the generation of pairs we study the case where the difference in the observation pairs is present in one random FoV. Whenever we sample the difference to be in one of the correlated FoV, its respective value in each pair is drawn from the probability distribution conditioned on the other correlated FoV. This means the difference in this factor is typically very small and depends on the correlation strength. Note that this procedure assures that constructed pairs are consistent with the observational data such that the correlation is never broken. Fig. 7 summarizes the weak supervision results when imposing correlations in object size and azimuth. We consistently observe much better disentangled models, often achieving perfect DCI score irrespective of correlations in the dataset. The latent spaces tend to strongly align their coordinates with the ground truth label axis. Finally, weak-supervision reduces unfairness relative to the unsupervised baseline, and occasionally achieves zero unfairness score.

These results suggest that weak supervision can provide a strong inductive bias capable of finding the right factorization and resolving spurious correlations for datasets of unknown degree of correlation. As a prominent example, this is an issue in the fairness context where real-world datasets often display unknown discriminatory correlations. Additional results on the other datasets can be found in Appendix B.2 including two additional scenarios where one has intervening capabilities to generate the pair. We consistently observe the same strong trends regarding disentangled correlations in all of the above studies using weak supervision.

## 5 CONCLUSION

We have presented the first large-scale empirical study examining how modern disentanglement learners cope with correlated observational data. We find that existing methods fail to learn disentangled representations of correlated factors of variation, and moreover that standard disentanglement metrics are insufficient to reveal these pairwise entanglements. We discussed practical implications for downstream tasks like fair decision making. Finally, we demonstrate how to correct for these latent correlations via various weakly supervised training scenarios. Our results thus support the importance and usefulness of learning independent mechanisms rather than independent factors of variations (Schölkopf, 2019; Parascandolo et al., 2018; Suter et al., 2019; Goyal et al., 2019). Besides the simple correlations studied in this work, future work is needed to address the open question whether these results extend to more complex nonlinear correlations and settings where many more variables are correlated simultaneously.

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

# A    IMPLEMENTATION DETAILS

**Unsupervised Disentanglement methods.**    For reasons of comparison, the considered disentanglement methods in this work cover the full collection of state-of-the-art approaches in `disentanglement_lib` from Locatello et al. (2019b) based on representations learned by VAEs. The set contains six different methods that enforce disentanglement of the representation by equipping the loss with different regularizers that aim at enforcing the special structure of the posterior aggregate encoder distribution. A detailed description of the regularizer forms used in this work, specifically $\beta$-VAE (Higgins et al., 2017a), FactorVAE (Kim & Mnih, 2018), AnnealedVAE (Burgess et al., 2018), DIP-VAE-I, DIP-VAE-II (Kumar et al., 2018) and $\beta$-TC-VAE (Chen et al., 2018) is provided in (Locatello et al., 2019b). We use the same encoder architecture with 10 latent dimensions for every model.

**Evaluation metrics.**    To measure disentanglement of a learned representation, various metrics have been proposed, each requiring access to the ground truth labels. The BetaVAE score is based upon the prediction of a fixed factor from the disentangled representation using a linear classifier (Higgins et al., 2017a). The FactorVAE score is intended to correct for some failures of the former by utilizing majority vote classifiers based on a normalized variance of each latent dimension (Kim & Mnih, 2018). The SAP score represents the mean distance between the classification errors of the two latent dimensions that are most predictable (Kumar et al., 2018). For the MIG score, one computes the mutual information between the latent representation and the ground truth factors and calculates the final score using a normalized gap between the two highest MI entries for each factor. Finally, a disentanglement score proposed by Eastwood & Williams (2018), often referred to as DCI score, is calculated from a dimension-wise entropy reflecting the usefulness of the dimension to predict a single factor of variation.

**Unfairness between a pair of FoV**: The scores reported are based on a notion of demographic parity for predicting a target variable $y$ given a protected and sensitive variable $s$. Both $y$ and $s$ can be associated with a factor of variation here. Rather than using the global total variation average as defined in Locatello et al. (2019a), we report the individual demographic parities for the correlated factors specifically.

**Pairwise entanglement metric**: In order to further analyze the differences between the GBT feature importance matrices between latent code and encoded factor of variation, we view them as weights on the edges of a bipartite graph encoding the statistical relation between each factor of variation and code. We can now delete all edges with weight smaller than some threshold and count (i) how many factors of variation are connected with at least a latent code and (ii) the number of connected components with size larger than one. We can then compute which factors are merged at which threshold. Factors that are merged at lower threshold are more entangled in the sense that are more statistically related to a shared latent dimension. This computation can be not only based on the GBT feature importances but likewise on weight matrices inferred from the Mutual Information.

**Joint distributions of correlated factors in datasets:**    In Fig. 8 we show the joint probability distributions of the correlated pair of FoV for all datasets and correlation strengths considered in this study. Dataset A, B and C were designed with correlated factors of variation that are ordinal for a natural visual interpretation of the traversals. In contrast, datasets D and E contain a correlated factor of variation that has no such natural ordering.

# B    ADDITIONAL RESULTS

## B.1    SECTION 3

**Shortcomings of existing metrics.**    Following recent studies, we evaluate the trained models with the help of a broad range of disentanglement metrics that aim at quantifying overall success by the help of a single scalar measure. Perhaps surprisingly, as can be seen in Fig. 9, there is no clear trend among all implemented disentanglement scores w.r.t. correlation strength. The metrics have been evaluated by sampling from the correlated data distribution although they do not differ substantially when evaluated on the uncorrelated distribution. We thus argue that commonly used methods are insufficient to provide insight into the latent space learned.

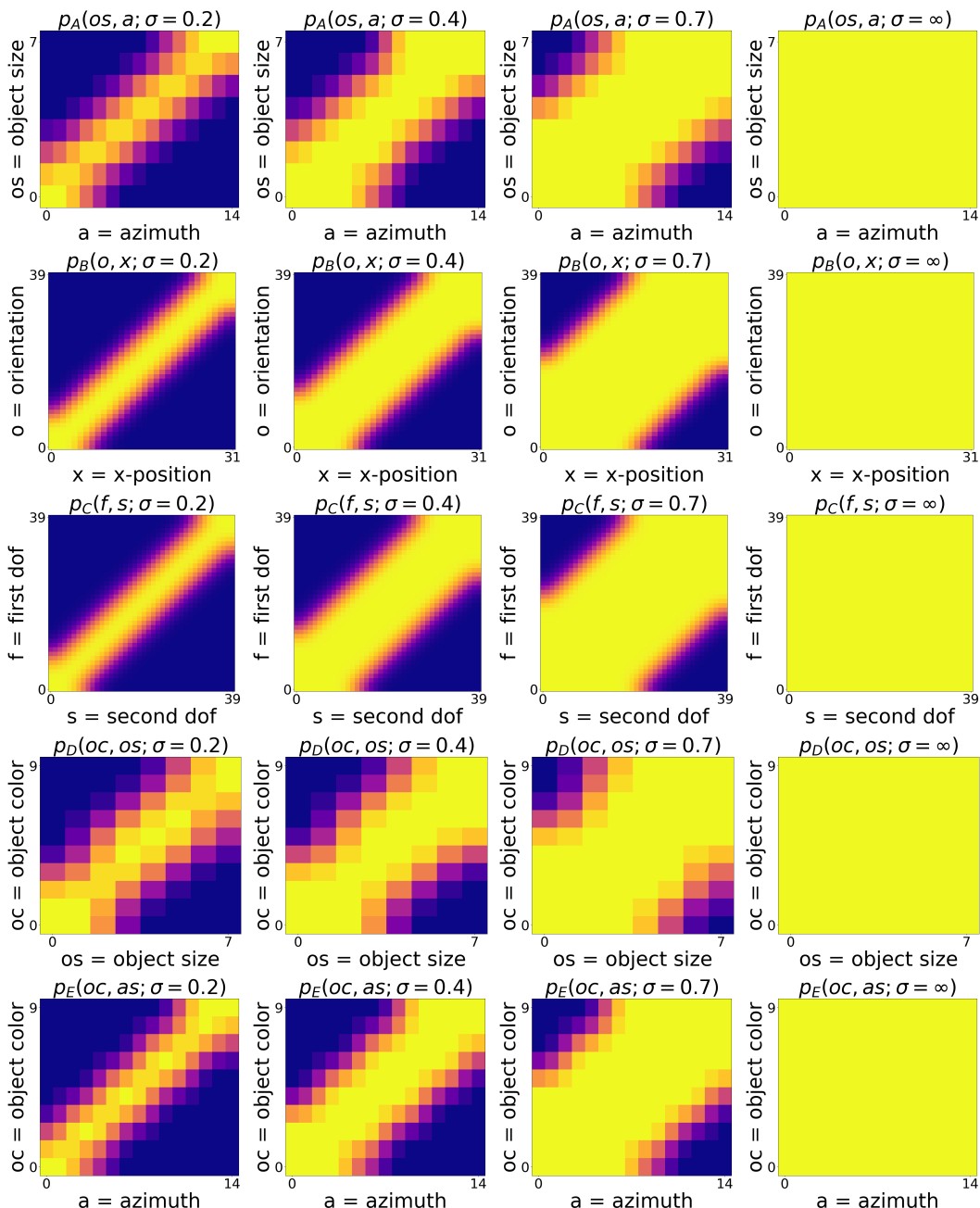

Figure 8: Probability distributions for sampling training data in the correlated pair of FoVs in the respective datasets (A, B, C, D, E) considering correlation strengths of $\sigma = 0.2$, $\sigma = 0.4$, $\sigma = 0.7$ and $\sigma = \infty$, the uncorrelated limit (from left to right).

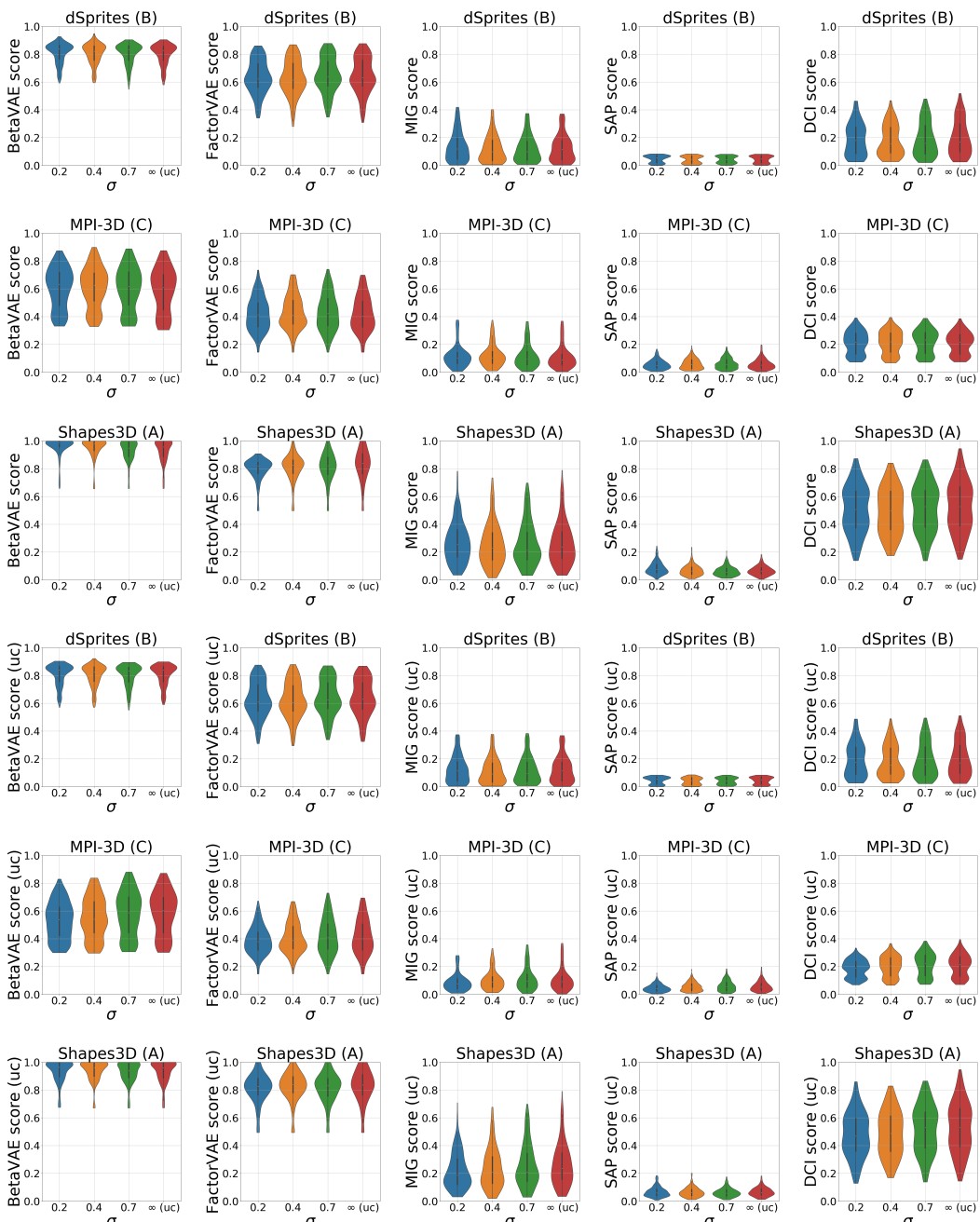

Figure 9: Standard disentanglement metrics evaluated on the correlated and uncorrelated (uc) training set showing no clear trend for different correlation strengths.

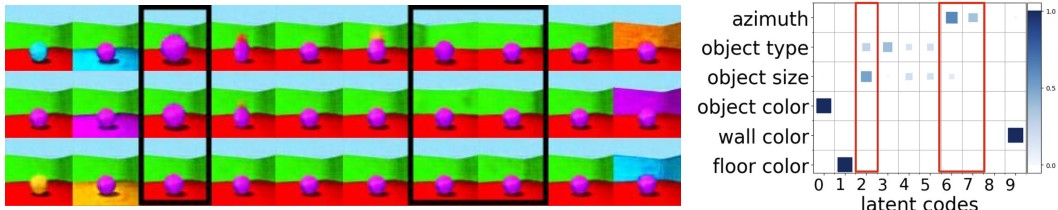

Figure 10: We show latent traversals (left) of the best DCI score model among all 180 trained models with weak correlation ($\sigma = 0.7$) in object size and azimuth. The traversals in latent code no 3 and 7/8 (highlighted in black), suggest that these dimensions encode no mixture of azimuth and object size compared to the models with stronger correlation. This is supported by the GBT feature importance matrix of this model (right).

| correlation strength $\sigma$ | | 0.2 | 0.4 | 0.7 | $\infty$ (uc) | 0.2 | 0.4 | 0.7 | $\infty$ (uc) |
|---|---|---|---|---|---|---|---|---|---|
| Shapes3D (A) | object size - azimuth | 0.38 | 0.26 | 0.13 | 0.08 | 0.28 | 0.25 | 0.2 | 0.17 |
| | median other pairs | 0.09 | 0.09 | 0.09 | 0.08 | 0.2 | 0.2 | 0.19 | 0.18 |
| dSprites (B) | orientation - position x | 0.17 | 0.16 | 0.14 | 0.11 | 0.34 | 0.31 | 0.24 | 0.14 |
| | median other pairs | 0.13 | 0.13 | 0.13 | 0.13 | 0.16 | 0.18 | 0.19 | 0.15 |
| MPI3D (C) | First DOF - Second DOF | 0.2 | 0.19 | 0.17 | 0.16 | 0.54 | 0.52 | 0.5 | 0.49 |
| | median other pairs | 0.16 | 0.16 | 0.15 | 0.15 | 0.25 | 0.25 | 0.26 | 0.25 |

| correlation strength $\sigma$ | | 0.2 | 0.4 | 0.2 | 0.4 |
|---|---|---|---|---|---|
| Shapes3D (D) | object color - object size | 0.29 | 0.28 | 0.38 | 0.31 |
| | median uncorrelated pairs | 0.07 | 0.07 | 0.11 | 0.11 |
| Shapes3D (E) | object color - azimuth | 0.25 | 0.23 | 0.43 | 0.3 |
| | median uncorrelated pairs | 0.1 | 0.09 | 0.15 | 0.15 |

Figure 11: Pairwise entanglement scores help to uncover still existent correlations in the latent representation. Left: Mean of the pairwise entanglement scores for the correlated pair (red) and the median of the uncorrelated pairs. We see that stronger correlation leads to statistically more entanglement latents across all datasets studied compared to their baseline pairwise entanglement where the data exhibits no correlations (blue). Each pairwise score is the mean across 180 models for each dataset and correlation strength. Scores in the left table are based on GBT feature importance and scores presented in the right table are based on Mutual Information.

**Latent structure and pairwise entanglement.** Our hypothesis that the latent representations are less correlated if the correlation strength is weaker is shown for a model on Shapes3D (A) with weak correlation in Fig. 10. Here the latent traversals do not mirror the major and minor axis of the correlated joint distribution. This conclusion is being backed by the thresholds of the pairwise entanglement metrics for the correlated pair vs. the median of all other pairs across all datasets, either when computing them using the GBT feature importances or the mutual information. See Fig. 11 for these respective results. Another pairwise metric that tracks the correlation strength in our scenario is the unfairness score between the correlated pair of factors that is being shown for datasets A, B and C in Fig. 12.

**Generalization Properties** In order to support our conclusion from these results that disentanglement methods can generalize towards unseen FoV configurations we show in Fig. 13 latent traversals originating from these OOD point with smallest object size and largest azimuth. We observe that changes in the remaining factors reliably yield the expected reconstructions. Additionally, we are interested in where samples from correlated models are located with respect to the ground truth value of each correlated variable. For this, we are visualizing the latent spaces with similar extrapolation and generalization capabilities of four models from the two strongest correlation dataset variants of Shapes3d (D) and Shapes3d (E) in Fig. 14.

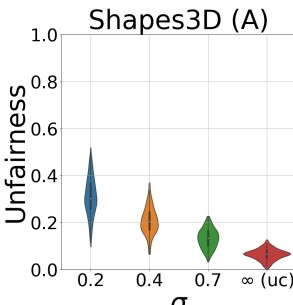 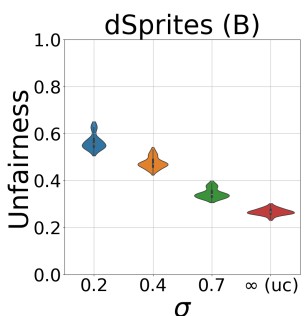 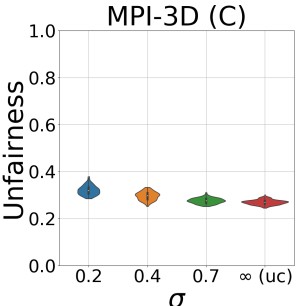

Figure 12: Disentangled representations trained on correlated data are anti-correlated with higher fairness properties. The plots show the mean unfairness scores between the correlated factors with decreasing correlation strength for Shapes3D (A), dSprites (B) and MPI3D-real (C).

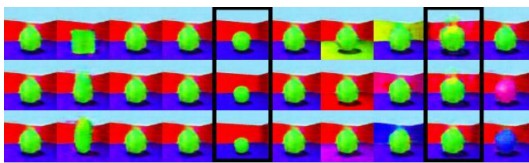

Figure 13: Generalization capabilities towards out-of-distribution test data. Latent traversals from an observations the model has never seen during training. The starting point corresponds to a factor configuration in point 1 from Fig. 5. Shown are the results of the model with highest DCI score among all 180 trained models on Shapes3d (A) with a very restricted correlation strength $\sigma = 0.2$ in object size and azimuth

## B.2 SECTION 4

**Post-hoc alignment correction with few labels** In Fig. 15, we see the axis alignment of the correlated latent space after fast adaptation using linear regression on a model trained on Shapes3D (A). Fast adaptation with linear regression substitution fails in some settings: when no two latent dimensions encode the applied correlation isolated from the other latent codes, or when the correlated variables do not have a unique natural ordering (e.g. color or categorical variables). Additionally, the functional form of the latent manifolds beyond the training distribution is unknown and in general expected to be nonlinear. We test the possibility of fast adaptation in this case using as substitution function a one-hidden layer MLP classifier of size 100 on the correlated Shapes3D variants. Under this method, we sample the FoV from a uniform independent distribution. A small number of such samples could practically be labeled manually. Using only 1000 labeled data points for our fast adaptation method shows a significant reduction in disentanglement-thresholds for the correlated pair (Fig. 16).

**Alignment during training using weak supervision** Using the studied weakly supervision Ada-GVAE method with $k = 1$ from Locatello et al. (2020), we showed that weak supervision can provide a strong inductive bias capable of finding the right factorization and resolving spurious correlations for datasets of unknown degree of correlation. Despite the results shown on Shapes3D (A) in the main paper, results across all three correlation variants in Shapes3D (A, D, E) are shown in Fig. 17 as well as some representative latent space visualizations that show strong axis alignment in Fig. 18. This study contains a total of 360 trained models.

In addition to the experiment from the main paper where pairs are constructed solely from the correlated observational data, we want to study two scenarios where we have limited intervention capabilities on the FoV to generate training pairs. The resulting distribution of FoVs (still exhibiting correlations) in these pairs depends on whether the correlation between two pairs is due to a causal link or due to a common confounder.

**Scenario I-1:** We assume there is a confounder causing a spurious correlation between the factors, such that correlation is broken whenever we constraint the change to be in one of the correlations.

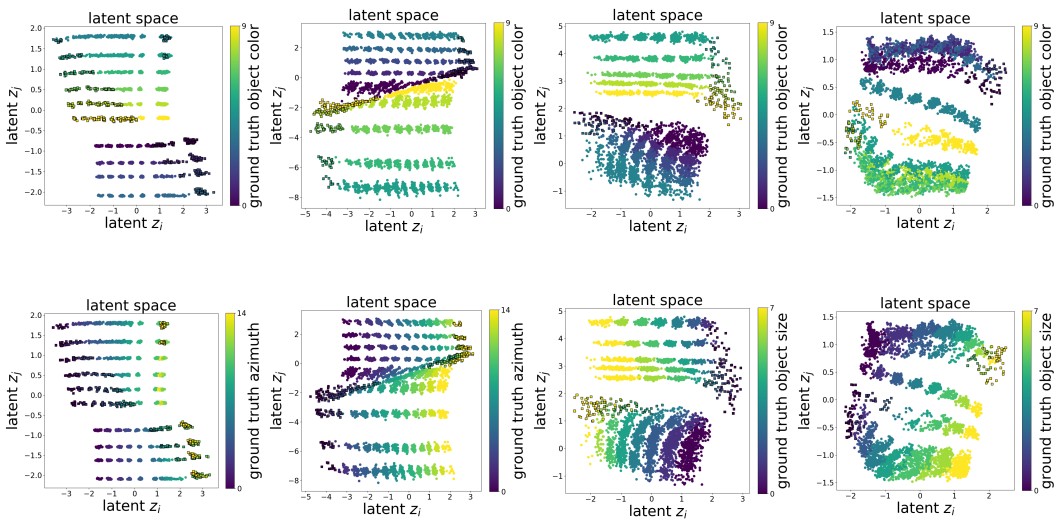

Figure 14: Latent space distribution of the two entangled dimensions of the best DCI model in Shapes3d (E) with $\sigma = 0.2$ (first column), in Shapes3d (E) with $\sigma = 0.4$ (second column), in Shapes3d (D) with $\sigma = 0.2$ (third column) and in Shapes3d (D) with $\sigma = 0.4$ (fourth column). Latent codes sampled from correlated observations (circle without edge) and (2) latent codes sampled with an object size-azimuth configuration not encountered during training(squares with black edge). Each column shows the ground truth values of the two correlated factors by color.

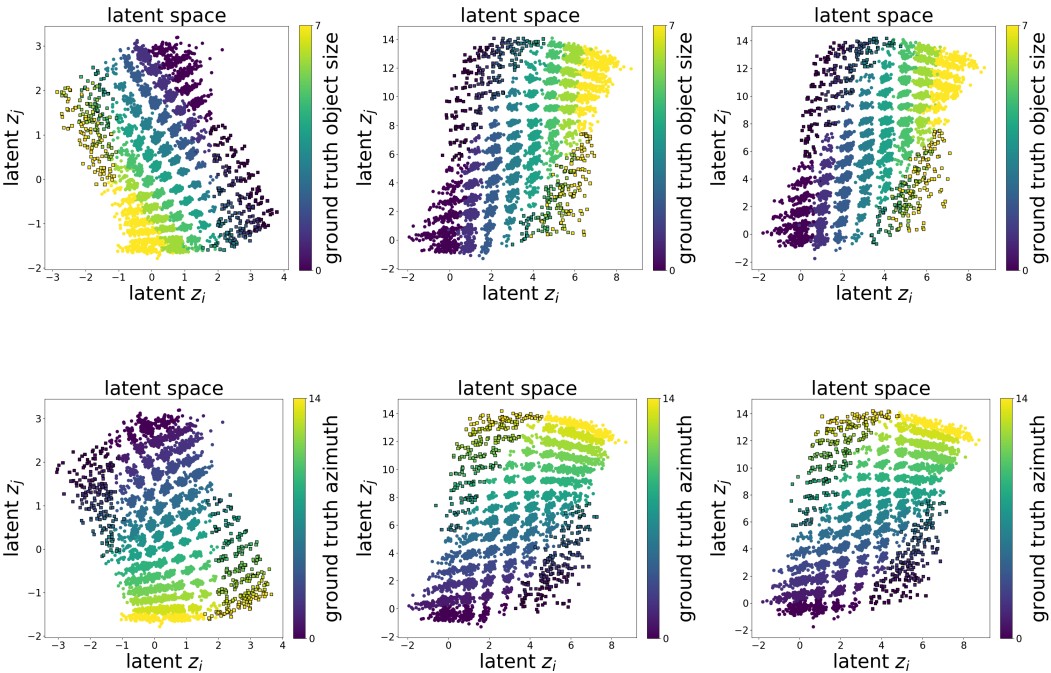

Figure 15: Latent space distribution of the two entangled dimensions of the best DCI model in Shapes3D (A). Latent codes sampled from correlated observations (circle without edge) and (2) latent codes sampled with an object size-azimuth configuration not encountered during training (squares with black edge). Left column shows the ground truth values of the two correlated factors by color. Middle and right column show the fast adapted space using linear regression and 100 or 1000 labels respectively.

| dataset | labels | 0 | 1000 |
|---|---|---|---|
| Shapes3D (D) $\sigma = 0.2$ | object color - object size | 0.3 | 0.16 |
| Shapes3D (D) $\sigma = 0.2$ | median uncorrelated pairs | 0.07 | 0.07 |
| Shapes3D (E) $\sigma = 0.2$ | object color - azimuth | 0.25 | 0.2 |
| Shapes3D (E) $\sigma = 0.2$ | median uncorrelated pairs | 0.1 | 0.11 |
| Shapes3D (A) $\sigma = 0.2$ | object size - azimuth | 0.37 | 0.26 |
| Shapes3D (A) $\sigma = 0.2$ | median uncorrelated pairs | 0.09 | 0.1 |

Figure 16: Mean of the pairwise entanglement scores for the correlated pair (red) and the median of the uncorrelated pairs (based on GBT feature importance) for all pairs of variables in Shapes3D (D) (top), Shapes3D (E) (middle) and Shapes3D (A) (bottom) all with correlation strength $\sigma = 0.2$. Each pairwise score is the mean across 180 models for each dataset and correlation strength. First column is the unsupervised baseline without any fast adaptation and the second column shows that fast adaption using a one-hidden layer MLP reduces these correlations with as little as 1000 labels when sampled from the uncorrelated dataset.

The value of the changing variable is then sampled uniformly in the second observation. Note that this still means that the vast majority of sampled pairs exhibit correlated FoV. This is depicted in the substantially lifted disentanglement scores shown in Fig. 19 as well as some selected latent space visualizations that show strong axis alignment in Fig. 20. We consistently observe much better disentangled models, often achieving perfect DCI score irrespective of correlations in the data set. The latent spaces tend to strongly align their coordinates with the ground truth label axis. We chose 10 random seeds per configuration in this study, yielding 720 models in total.

**Scenario I-2:** Let us assume $C_1$ causes $C_2$ in our examples, which manifests as the studied linear correlations. Within this setting we cannot sample uniformly in $C_2$ if we intervene (or "fix") all factors except for this causal factor. Intervening on all factors but $C_1$, however, allows us to sample any value in $C_1$ as it is not causally affected by $C_2$. To test the hypothesis that this constraint still allows for disentangling the correlation, we trained on Shapes3D (A) and sample pairs consistent with this causal model. Besides observing visually disentangled factors in the latent traversals, we show a summary of our results in Fig. 21 with the same significant improvements regarding disentangling the correlated FoVs. Besides the above correlation strengths, we additionally trained the same models using a very strong correlation of $\sigma = 0.1$, yielding 300 models trained in this study.

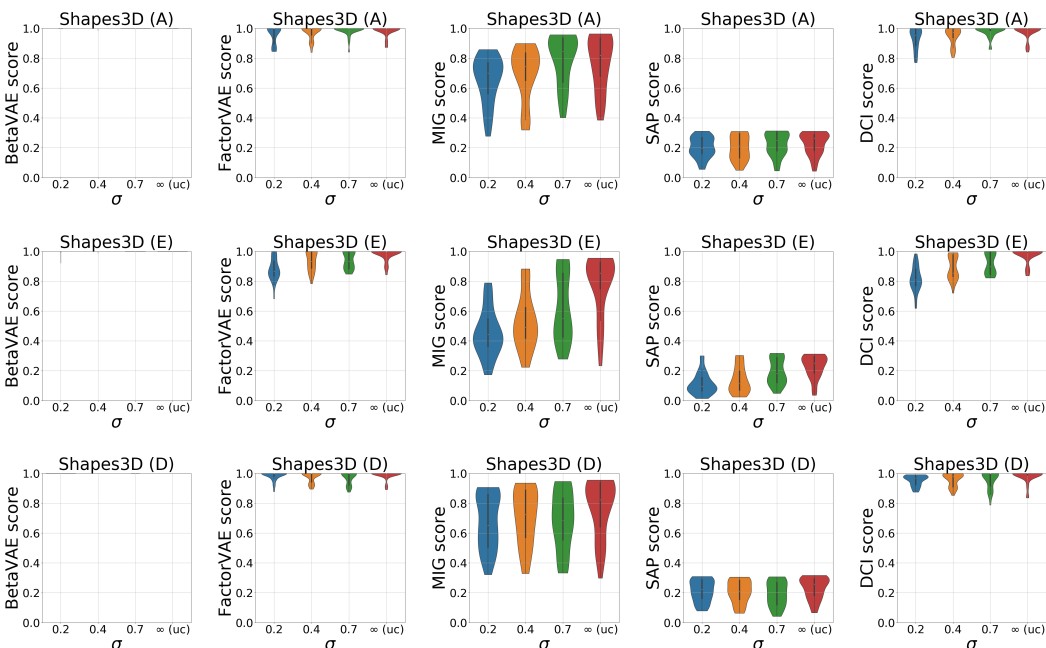

Figure 17: Standard disentanglement metrics evaluated for the weakly supervised scenario using correlated observational data.

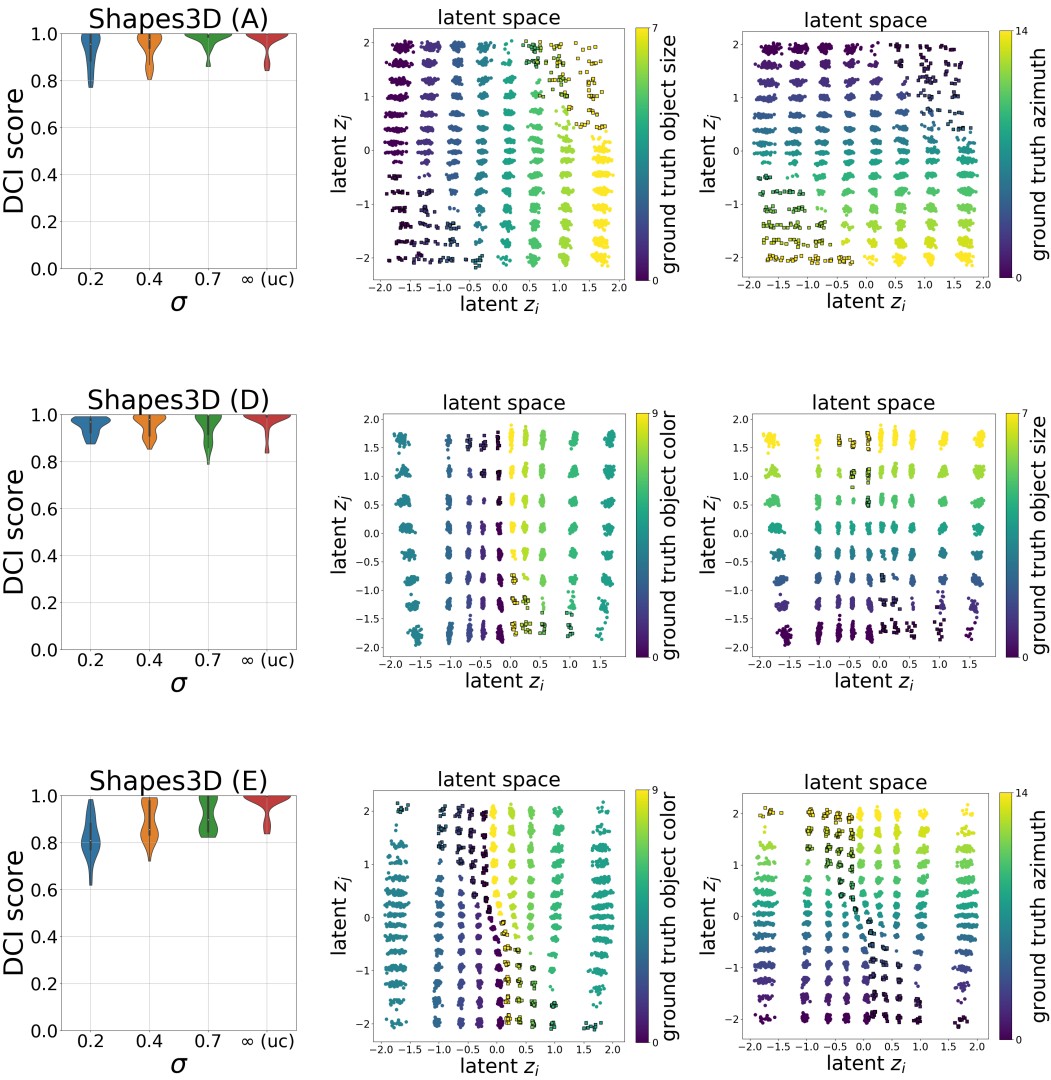

Figure 18: Left: For the weakly supervised scenario using correlated observational data trained models on Shapes3D (A), (D) and (E) correlating object color and azimuth learn consistently improved, often perfect, disentangled representation across all correlation strengths. Right: Latent dimensions of a best DCI model trained on strongly correlated observational data. Representations are perfectly axis-aligned with respect to both of the correlated variables ground truth values (right).

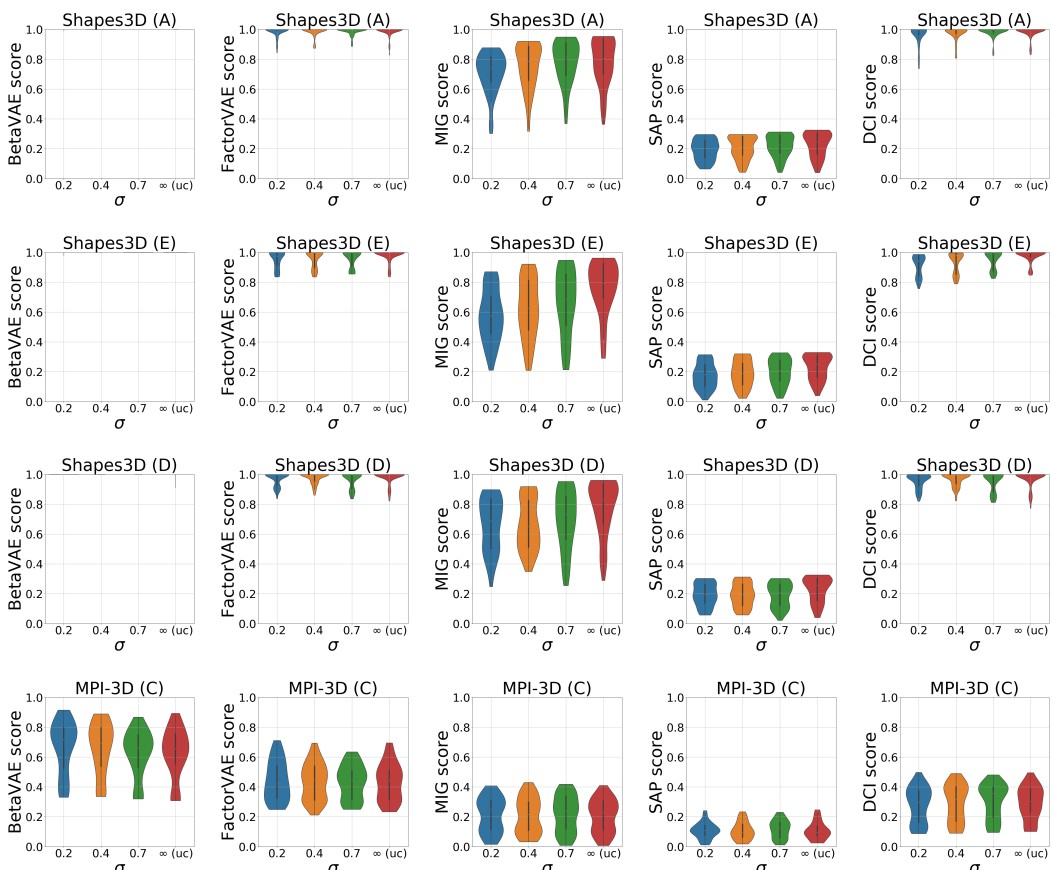

Figure 19: Standard disentanglement metrics evaluated on the correlated training sets for the weakly supervised scenario with intervening capabilities (I-1).

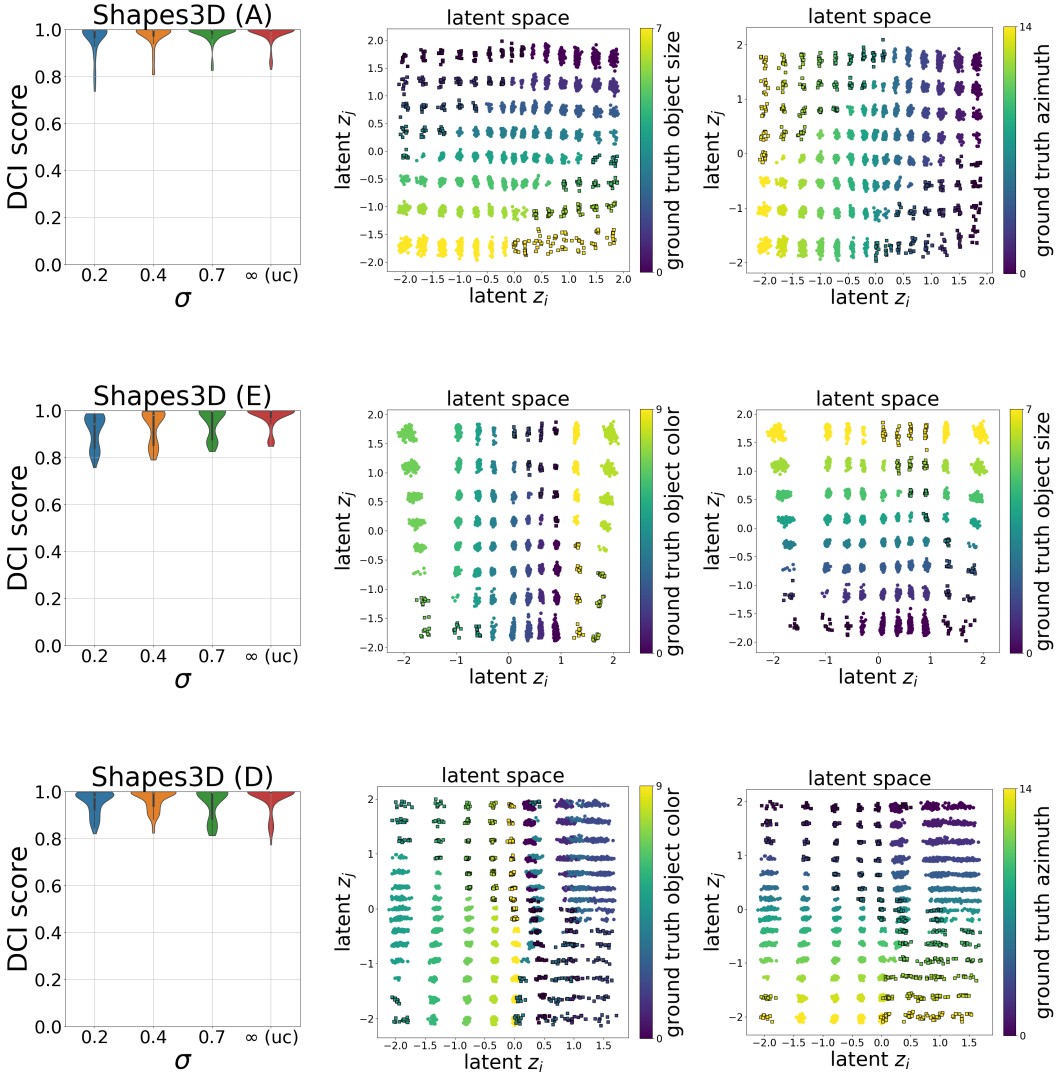

Figure 20: Left: For the weakly supervised scenario with intervening capabilities (I-1) trained models on Shapes3D (A), (D) and (E) correlating object color and azimuth learn consistently improved, often perfect, disentangled representation across all correlation strengths. Right: Latent dimensions of a best DCI model with strong correlation (0.2). Representations are perfectly axis-aligned with respect to both of the correlated variables ground truth values (right).

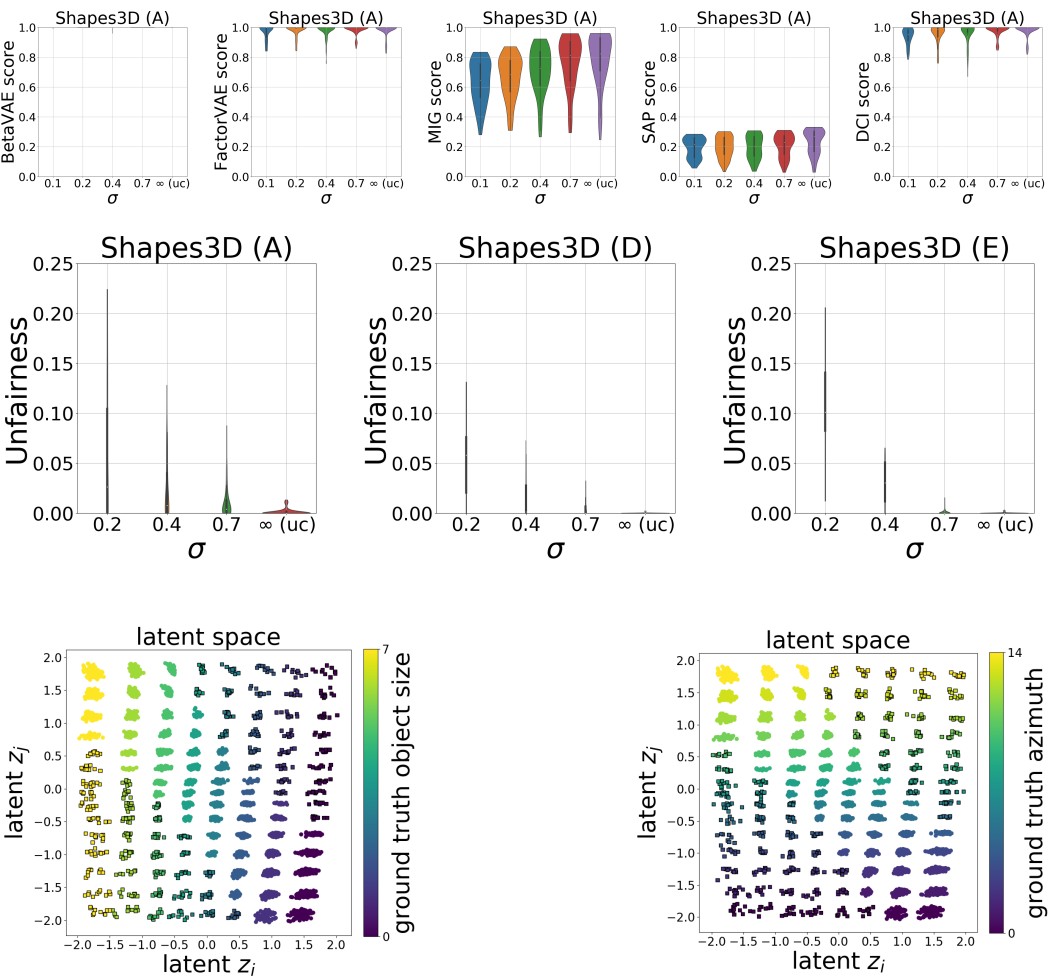

Figure 21: Disentanglement metrics (top), unfairness scores (middle) and latent spaces (bottom) show strong disentanglement using weak supervision with intervening capabilities (I-2) - even under the stronger assumption that sampling of observation pairs follow its causal generative model. We show the learnt latent space encoding of the two correlated factors of variation for a model with $\sigma = 0.1$ and low reconstruction loss.

