# OpenReview forum: "On Disentangled Representations Learned From Correlated Data"
_ICLR.cc/2021/Conference — Reject_

### Official Review · AnonReviewer4 · 2020-10-26
**Evaluation is extensive, but the conclusions drawn from them are either 1. not justified very well by the evaluation 2. unsurprising/already known.**

**Rating:** 6
**Confidence:** 5

**Review:**

The paper studies the behaviour of disentanglement methods and metrics on data where a couple of factors of variation (FoV) are correlated, a more realistic setup compared to the usual independent FoV setting in the literature. The paper shows how the correlation in the FoV is reflected in the representations learned by the models, and claims that the widely used disentanglement scores fail to capture these correlations. A couple of solutions that use weak supervision are suggested.

Strengths:
1. The paper attempts to address an important limitation of current unsupervised disentangling methods, in that they assume independent FoV. I agree that this is an unrealistic assumption, and only holds for the synthetic datasets used in the literature.
2. The experimental evaluation of the paper is extensive
3. The paper is presented clearly for the most part.

Weaknesses:
1. At the beginning of Section 3.2, the claim that the existing disentanglement metrics fail to capture correlations in the training data seems flawed. Figure 2, that the violin plots of metrics for 180 models (with different loss/hyperparam combinations/random seed) shows no clear trend, is used as evidence for the claim. However this argument ignores an important confounding factor, which is the behaviour of the models (trained with losses originally designed for data with independent FoV) when trained on a dataset with correlated FoV. Without understanding the behaviour of models after training, it is difficult to draw any conclusions from Figure 2. Moreover comparing violin plots of all 180 models does not seem sensible - the models with poor disentangling performance will likely be uninformative, creating noise for comparisons between different settings. Hence I think it makes more sense to compare results for the top models for a given metric (as is done in Fig3).
2. I also have doubts regarding the conclusion itself, that all existing metrics fail to capture correlations in the training data. I think this only holds for some metrics and not for others. Suppose the representations that we are evaluating are the ground truth FoV values. Then the BetaVAE and FactorVAE metrics will give a perfect score regardless of the degree of correlation by design. But the MIG/SAP/DCI scores will be lower for higher correlation in the FoV, since the mutual information/prediction accuracy between the two correlated factors will be higher than between uncorrelated factors.
3. The result presented by Figure 3, that training disentanglement models on strongly correlated data gives representations encoding mixtures of the correlated FoV, is unsurprising in my opinion, and adds little value to the pool of knowledge in the literature.
4. In Section 2, I think that the statement “it has been shown that purely unsupervised learning of disentangled representations is impossible (Locatello et al., 2019b)” is a wrong interpretation of Locatello et al. They show that optimising marginal likelihood in a generative model (such as a VAE) cannot achieve disentangling without any inductive biases in the model. But the inductive biases in the models used by disentangling methods, along with the loss (that is a variant of the ELBO and not the marginal likelihood), are what allow disentangling in practice. There are also theoretical works such as [1] that explain this behaviour.
5. In Section 3.2, the conclusion “These results suggest that we cannot expect disentangled representations learned unsupervisedly to help reduce unfairness beyond the benefits discussed in Locatello et al. (2019a)” seems problematic. It’s not the unsupervised learning that’s problematic, rather it’s the model assumption that the FoV are independent that is problematic in the scenario of correlated FoV, which is an unsurprising conclusion.
6. The conclusion in Section 3.3, that disentanglement methods can generalize towards unseen FoV configurations, is old news that is already shown in the literature in disentanglement models trained on datasets with an incomplete set of FoV configurations e.g. 3D Faces, CelebA
7. In Section 4.2, the claim “This kind of extra knowledge is often available at no cost , e.g., in temporarily close frames from a video of a moving robot arm where some factors remain unchanged” is unjustified. We still need to know the number of factors that have changed, which usually requires human labelling
8. The weak supervision for the correlated FoV when applying Ada-GVAE relies on being able to generate data where the correlation is broken, which seems like an unrealistic assumption. The authors seem to address this by assuming a causal relation between the two factors. However even if there is a causal relation between the two factors (C1 causes C2), the correlation implies that some (C1,C2) pairs are very unlikely to appear in the data, so you cannot “sample any value in C1” given C2. Hence I don’t understand how the causal setting at the end of Section 4.2 helps address the problems of having to generate data where the correlation is broken.

Overall the paper does address an important problem in the disentanglement literature, but the conclusions drawn from the extensive evaluation are either unsurprising or unjustified. Moreover, the proposed solution via weak supervision appears flawed because it requires generating data where the correlation is broken, a very unrealistic assumption.

Other points:
- Typo in Figure 2 caption: “lower \sigma indicates less correlation” <- “higher \sigma indicates less correlation”
- Section 3.1: “P(z_c1, z_c2) ~ N(z_c2 - \alpha z_c1, \sigma)” How does the RHS define a joint distribution? The RHS shows a scalar normal distribution, whereas the LHS is a joint density. Do you want to replace “\sim” with “ \propto” ?
- Why doesn’t entanglement decrease with #labels > 100 in Figure 6?
- It might be helpful to also look at the disentanglement metrics for just the two correlated factors, to further highlight the differences between different models.

[1] Rolinek, M., Zietlow, D. and Martius, G., Variational Autoencoders Pursue PCA Directions (by Accident). CVPR 2019.

===========================

Score raised to 5 following response to the rebuttal below, then to 6 following the re-rebuttal.

---

> ### Author Response · Authors · 2020-11-13
> **Response to AnonReviewer4**
>
> We would like to thank the reviewers very much for the extensive review and useful comments. In the following we would like to address your comments hoping they will clarify raised concerns:
>
> 1.) We agree that conclusion from looking at the violin plots only without understanding the behavior of the models can be misleading, but in the remainder of this section we extensively study this on a pairwise level. When utilizing these insights we can draw the conclusion about usefulness of the metrics in hindsight. We tried to clarify this in the updated version.
>
> 2.) We agree that BetaVAE and FactorVAE will give a perfect score if you would take the ground truth correlated representations for their computation. Thank you for pointing out this fact. We updated the paragraph accordingly. As we see in many of our trained models, the BetaVAE score is showing perfect disentanglement even if the latent space is rotated, i.e. the model in Fig 3 has beta score 1 but traversals are entangled. This should be an argument in support of our conclusion. The other scores measure MI and prediction accuracy in the latent codes but as can be seen from Fig 10/11 they do not show this trend along different correlation strengths in our setting. Still, we see these persisting correlations in the pairwise metrics.
>
> 3.) Although the fact that models learn representations that encode mixtures of the correlated FoV might be unsurprising we are not aware of literature that showed this in a large-scale empirical study across a wide range of disentanglement methods. Therefore we considered it important to investigate this as it (a) validates these intuitions and (b) bases this conclusion on extensive experimental results.
>
> 4.) We agree that we missed mentioning the inductive bias aspect in the interpretation of the non-identifiability result. Thanks for pointing that out. We updated that in our paper. In practice, however, the biases of the disentangling methods are insufficient to reliably produce disentangled representations (see Fig 2, 4 center, 5 in the original ICML paper).
>
> 5.) The objective of many disentanglement learners is to learn independent explanatory FoV. Locatello et al. (2019a) showed that this is beneficial in terms of fairness with respect to a protected variable in the independent setting. In fairness settings, however, the target and sensitive variables are typically correlated. If we expect disentanglement learners to learn an independent factorization of correlated FoV then this should be helpful for fairness. We agree that it is true that the conclusion might seem unsurprising but only once you know that the disentangled learners cannot disentangle this correlation. Additionally,  we show that this notion of fairness based on a total variation measure is another pairwise metric to uncover the remaining correlations in the latent space.
>
> 6.) While we agree that there are some interesting models that indicate generalization to factor combinations on incomplete sets, the true FoVs are generally unknown. We do not want to claim novelty in this observation but consider this behavior of the models an important insight as we do not think this kind of generalization towards unseen FoV configurations with zero probability in the prior is being generally considered solved by the community. We want to note that reviewer 2 finds this behavior interesting.
>
> 7.) We decided to investigate this particular method of weak supervision in our correlated setting as an example due to its recent popularity within the line of methods that learn disentanglement with identifiability guarantees. It is true that this form of weak supervision might require some form of human labeling but only knowing that some factors differ without knowing which specifically as well as their labels is substantially weaker. We guess it is also worth noting that Locatello showed this method is likewise working for k>1 or k=random, which seems natural to assume for temporary close frames. However, we understand that this might be not easy to achieve in every setting why we decided to tone down the corresponding claim.
>
> 8.) If there is a causal relationship between C1 and C2, specifically C1 causes C2, then it is true that it is very unlikely to observe some (C1, C2) pairs in the observational distribution. However, this is not the case anymore if one can intervene upon the variables, thereby generating samples from an interventional distribution. So if we intervene on C2, the causal link is removed and C1 is independent. We want to emphasize that the vast majority of pairs in this setting still exhibits these correlations and only in rare cases (where k corresponds to C2) we will sample some of these pairs.
>
> 9.) Why doesn’t entanglement decrease with #labels>100 in Fig6?
> Linear regression only needs a small number of labels to learn the optimal parameter for rotation and any additional labels will have no substantial additional benefit. See Fig16 for visualization.

---

> > ### Author Response · Authors · 2020-11-16
> > **Additional experiments weak supervision method**
> >
> > Following the reviewers’ concerns regarding the applicability of the weakly-supervised method in the original two scenarios where, in rare cases, an image contains some (C1, C2) that are unlikely to appear in observational data: We added a third scenario to the paper if we have no such intervening capabilities (see Appendix B.2). In this experiment, the two images of a pair are solely constructed from the observational correlated distribution without any otherwise unlikely (C1, C2) combinations. This analysis comprises 150 additionally trained models with similarly high disentangled latent spaces concerning the two correlated FoV. We want to thank the reviewer for this helpful comment that allowed us to improve this section.

---

> > > ### Comment · AnonReviewer4 · 2020-11-17
> > > **Reviewer response**
> > >
> > > **Response to 1)**
> > > Thinking about it more carefully I think the claim that existing disentanglement metrics are limited for capturing correlations is more justified than I had previously thought. Let me summarise to make sure I have understood correctly:
> > > - the rest of the section shows how the different disentangling methods give rise to entangled representations for the strongly correlated case (low sigma) and disentangled representation for the weakly correlated case.
> > > - The pairwise entanglement score quantifies this phenomenon when averaging across all models for a given sigma, and Figure 3 is a qualitative demonstration for the best performing model.
> > > - In this context, the proposed argument is that these differences are not captured by the disentanglement metric, and the evidence is that in Figure 2, there are no clear trend in the metric scores across all models for different values of sigma.
> > >
> > > However I still think that a limitation of this argument is that it's based on a violin plot across all 180 models instead of for the single best model for a given metric. As mentioned in my review, the models with poor disentangling performance will likely be uninformative, creating noise for comparisons between different settings. And usually when we want to obtain disentangled representations with a given method and metric, we would pick a single model with the top score by the metric. Hence I think it makes sense to show the result for just the single best model for each sigma and metric. The conclusion may be the same, but this would be more convincing evidence.
> > >
> > > **Response to 4)**
> > >  I see that you have added "without further inductive bias" to the sentence, but this sentence is still misleading/ambiguous. I think "Purely unsupervised learning of disentangled representations" should be replaced with something more specific such as "unsupervised disentangling by optimising marginal likelihood in a generative model". Same goes for the first sentence of Section 4.2.
> > >
> > > **Response to 8)**
> > > In the original draft, the method assumed that this intervention is possible, i.e. that you can generate data where the correlation is broken, and this appeared to be the biggest limitation of the proposed method. In your most recent comment about new experiments in Appendix B.2, it seems that this assumption is not strictly necessary, which would substantially improve the paper. But this would also mean that Section 4.2 should be heavily modified and you would need to reproduce Figures 6 & 7 for the new methods.
> > >
> > > **Overall response**
> > > While I still think that the results and conclusions shown in the paper are unsurprising, the element of surprise is a nice-to-have rather than a necessary criterion for evaluating a paper. Arguably the more important criterion is correctness, for which I think this paper does a decent job given the clarifications. Hence I raise my score to 5, but I do think that the submission can be made stronger in a future submission when Section 4.2 is modified to show results for the setting without any intervention.

---

> > > > ### Author Response · Authors · 2020-11-20
> > > > **Thanks for your feedback**
> > > >
> > > > Thanks for your reply and additional feedback. We greatly appreciate that you took our clarifications and improvements into account.
> > > >
> > > > Response to 1) We are happy to see that you agree with our conclusion now and want to confirm that your summary about how we came to these conclusions is correct. In our view, basing the evaluation on the top models would be correct if we consider semi-supervised approaches. However, the computation of the metrics requires ground truth labels, why model selection for the best models should ideally not be made using these test metrics. Therefore, model selection should be based on some unsupervised heuristic, but this choice might significantly bias the study and its conclusions.
> > > >
> > > > Response to 4)  We’d like to thank you for the alternative formulation and updated it accordingly in sections 2 and 4. Please see the updated version.
> > > >
> > > > Response to 8) We are happy to see that you think the new experiments would substantially improve this paper. Therefore, encouraged by your suggestions, we ran additional experiments in this setting without any interventions on further correlated dataset variants. This comprises 360 additional models compared to the original submission. Our results confirm that the assumption about intervening capabilities is not strictly necessary for the method to work. We still consistently observe much more disentangled models, often achieving perfect DCI score irrespective of the correlated observational training data. The latent spaces tend to strongly align their coordinates with the ground truth label axis. Following your proposal, we now modified section 4.2 based on these experiments and we report these results in Fig. 7. We moved the experiments with interventions to the appendix for the interested reader. Please see the updated draft.
> > > >
> > > > If there is anything else that we have not addressed regarding your concerns, we'd be happy to take another look. We very much appreciate your time and efforts with this review.

---

> > > > > ### Comment · AnonReviewer4 · 2020-11-20
> > > > > **Reviewer response 2**
> > > > >
> > > > > I've read the updated Section 4.2 and I believe my concerns for this section have been addressed. I will thus raise my score further to 6. My only concern is that the updates made to this section is fairly major compared to the original draft, so I'm not sure whether it aligns with ICLR reviewing rules. The area chair may have a better idea of this.

---

> > > > > > ### Author Response · Authors · 2020-11-20
> > > > > > **Thanks for your feedback**
> > > > > >
> > > > > > Thanks for your prompt reply. We appreciate your time and feedback very much and wanted to clarify that we did not remove any experiments or changed any conclusions but just moved some of the results to the appendix in light of your suggestions. If this would be a problem, we are happy to move the intervening experiments back to the main paper using page 9.
> > > > > >
> > > > > > If there is anything else that you think can improve the paper even further, we’d be happy to take another look.

---

### Official Review · AnonReviewer2 · 2020-10-29
**Timely empirical contribution to distangled representation learning with correlated factors of variation**

**Rating:** 7
**Confidence:** 3

**Review:**

This paper addresses the important problem that most existing disentangled representation learning algorithms analyze statistical independence rather than causal independence. The paper conducts a large scale study to investigate whether statistical correlation prevents learning disentangled representations (according to human defined ground truth factors of variation).

Pro:

This is a timely contribution to clarify problems in evaluating disentangled representation learning algorithms. The paper argues that assuming statistical independence is unrealistic, in practice factors of variation are often causally independent but statistically correlated. This seems like an important missing piece in current empirical evaluation benchmarks.

There are several interesting observations:

1.The existing metrics for disentanglement do not apply to situations with statistical correlation between the factors of variation. New metrics will be needed.

2.For correlated factors of variation, even though disentanglement fails, the latent space still extrapolates to never seen combinations.

3.Several semi-supervised or weakly supervised methods for disentanglement work well when the factors of variation are correlated.

Con:

Section 3 and 4 are a little hard to follow. There are many results with no clear logical connection to each other. Maybe it’s better to have some bullet points of the empirical findings, then point to specific paragraphs that explain the empirical methodology that produced these findings.

---

> ### Author Response · Authors · 2020-11-16
> **Response to AnonReviewer2**
>
> We want to thank the reviewers very much for their detailed review and helpful feedback, including suggestions on improving the structure of sections 3 and 4. We agree that a summary of the findings and conclusions in the beginning might help guide the reader through our analysis. Please see the updated draft, where we attempted to incorporate these suggestions.

---

### Official Review · AnonReviewer3 · 2020-10-30
**ON DISENTANGLED REPRESENTATIONS LEARNED FROM CORRELATED DATA**

**Rating:** 3
**Confidence:** 4

**Review:**

Summary: This paper systematically presents a large-scale empirical study on the disentangled representation learning when the underlying factors are possibly entangled. From the results of purely unsupervised settings, the authors have discovered the shortcomings of the existing metrics of disentanglement as well as the poor learned representations (in terms of disentanglement). However, with the help of small amount of factor labels or other weak supervision signals, recent approaches could learn fairly perfect representation.

First of all, it is worthy to pay attention to the possible correlation between factors when you intend to learning a disentangled representation. And the whole empirical results are carried out by very large number of experimental batches, which to some extent could well support the conclusions displayed in the paper.
However, there still exist several limitations in my opinion:
(1)	The design of induced correlation is too simple. As you have mentioned in related work, there were some papers noticed the too ideal assumptions of traditional VAE-based models which may not be held in practice. IMO, the linear dependency between only two variables is far from reality as well. More complicated settings should be involved.
(2)	In line with the former limitation, diagnostics of the potential entanglement should also not be limited to pairwise level, which cannot scale up to high dimensional latent factors.
(3)	The novelty of Section 4 is somewhat limited as all the correction methods and even some conclusions were proposed by the previous work.

---

> ### Author Response · Authors · 2020-11-13
> **Response to AnonReviewer3**
>
> We want to thank the reviewers for the time and effort spent on reviewing this paper. We appreciate your comment that we studied a problem worth paying attention to and the large number of experimental studies could support our conclusions. We feel the reviewers’ main concerns might be due to some degree of misunderstanding of the main contributions and novelty of such an extensive empirical study on inductive data biases on disentanglement learners with implications on fairness in particular and want to address your main concerns in the following.
>
> Concern (1): ‘The design of induced correlation is too simple. As you have mentioned in related work, there were some papers that noticed the too ideal assumptions of traditional VAE-based models which may not be held in practice. IMO, the linear dependency between only two variables is far from reality as well. More complicated settings should be involved.’
>
> We would like to emphasize that our primary goal and contribution is to study the behavior of disentanglement learners on correlated data systematically. Even though we acknowledge the fact that there is work done that indicates the need to study this problem, it is typically addressed in much smaller experiments not covering as many different methods (as detailed in the related work). This does not allow for drawing broader conclusions about this problem. We study this setting along with multiple datasets, multiple SOTA disentanglement learners, and random seeds. Additionally, we cover a systematic study with complete control and knowledge of the correlations present.
>
> Regarding more complicated settings: With a limited compute budget in mind, we decided to base this study on pairwise linear correlation as we can intuitively control and understand the correlation and also use this understanding for subsequent analysis of the latent space. This is already leading to an extensive number of models as is being acknowledged in the beginning of your review. Linear correlations are an important first step in studying this problem, and it already indicates several shortcomings, as discussed in the paper. It is not straightforward to come up with more complicated settings as the correlation variations would grow exponentially, e.g. when studying multiple variable correlations. Additionally, it is unclear how one could systematically analyze and understand the latent spaces of models in these (potentially even nonlinear) scenarios. Moreover, the compute required for the experiments is already significant (0.72 GPU years) and a tradeoff regarding additional insights should be taken into account. We are happy to add a more concrete experiment with nonlinear settings but would kindly ask for a concrete proposal (feasible within the rebuttal time-period) since the multiple variable nonlinear setting is too huge as a search space.
>
> Concern (2): ‘In line with the former limitation, diagnostics of the potential entanglement should also not be limited to pairwise level, which cannot scale up to high dimensional latent factors.’
>
> When studying settings with pairwise correlations in the ground truth factor, it makes sense to study the latent space on a pairwise level to obtain fine-grained insight into corresponding latent spaces. Additionally, we do not only study pairwise metrics but also five commonly used disentanglement scores utilizing the full latent space. We would appreciate, if the reviewers could elaborate on what they mean by ‘scale up to high dimensional latent factors’? Disentanglement learning aims to learn a *low-dimensional* set of explanatory factors of variation of high-dimensional (observational) data.
>
> Concern (3): ‘The novelty of Section 4 is somewhat limited as all the correction methods and even some conclusions were proposed by the previous work.’
>
> We want to point out that our primary goal is to validate the applicability of a significant line of work on disentanglement to more realistic scenarios where factors of variation are not independent. That is why we cover not only the unsupervised setting but also a popular weakly supervised method in disentanglement learning. Regarding the correction methods investigated in section 4: We are not aware of other work studying the fast adaptation method on pairs of entangled latent factors (section 4.1). The original weakly supervised method (section 4.2) was only investigated on perfectly independent priors, so it is not true that these conclusions are known.

---

> > ### Author Response · Authors · 2020-11-20
> > **The paper has been updated with additional results. Is there anything else you would like us to respond to?**
> >
> > We thank the reviewer for their feedback and valuable comments and wanted to let you know that we updated the paper in the meantime, following other reviewers' comments.
> > For the weakly-supervised method in section 4.2: We now report results for a third pair sampling scenario, which shows that this method works even under the weaker assumption without any intervening capabilities but with training pairs from the observational correlated distribution only. This includes 360 additionally trained models. Some of the previous results from the other two scenarios have now been moved to the appendix. Please see the updated paper. We would be happy to know if your concerns might have been resolved with these modifications?
> >
> > Now that the first phase of the response period is over, if you have time and could indicate if there are any other concerns of yours that we have not addressed, we'd be happy to take a look.
> >
> > Thanks for your time.

---

### Decision · Program_Chairs · 2021-01-07
**Final Decision**

**Decision:**

Reject

**Comment:**

This submission considers the problem of learning disentangled representations from data in which there are correlations between underlying factors of variation (FoVs). Much of the work on learning disentangled representations has considered simulated datasets in which the FoVs are conditionally independent. The authors perform an extensive experimental evaluation (4000+ trained models). The main findings of this evaluation are that:

- Existing methods fail to disentangle when ground truth FoVs are correlated, in the sense that learned factors will reflect the correlations in the training data
- Metrics for disentanglement do not necessarily reveal correlations in underlying factors
- Semi-supervision and weak supervision can be used to induce learned factors that align with true FoVs

Reviewers expressed diverging opinions on this paper:

- R2 is a favor of acceptance, but does note that the paper is somewhat difficult to follow, owing to the fact that it presents a large number of results and does not quite arrive at a streamlined narrative.

- R4 was initially critical and posted detailed comments relating to framing, interpretation of existing work, and the conclusions that can be drawn from the presented experiments. The authors were able to address a number of these concerns in a detailed discussion with the reviewer.

- R3 is critical of the experimental setup, which considers linear correlations between two underlying factors, and feels the semi-supervised and weakly supervised experiments have limited novelty. This reviewer did not respond to author feedback.

The metareviewer has carefully read the reviews, author feedback, and subsequent discussion. Owing to the fact that reviews are diverging the meteareviewer also read the paper.

As R4 notes, the results in this paper are in some sense unsurprising – we would expect correlations between underlying factors to lead to correlated learned factors. In fact one could even argue that dimensions in the latent space should reflect correlations in the training data. That said, the metareviewer feels that a paper need not present results that are surprising, as long as the experimental evaluation is rigorous and there are no major problems with framing and exposition.

In this context, the metareviewer would like to express their appreciation to R4 for taking the time to follow up in detail with the authors, and for checking their revisions. The metareviewer feels that the fact that these revisions have a fairly large edit distance should not in itself not impede acceptance, as long as reviewers are agree that the edits improve the paper.

The metareviewer is not entirely convinced by the criticisms presented by R3. The reviewer is of course correct that real-world datasets will not just have linear correlations between two factors. That said, an experimental evaluation of how correlations affect the degree of disentanglement has to start somewhere, and even these comparatively simple experiments represent a substantial effort on the part of the authors.

Having read the submission, the metareviewer agrees with R2s assessment that the exposition is difficult to follow, even for readers who know the relevant literature. As the reviewer notes, the overall narrative could be clearer. Extracting a clear narrative is challenging when there are many experiments to report, but it is nonetheless something that the authors should spend additional time and thought on. Another factor that hurts clarity is that experiments are described in long paragraphs that often would benefit from an equation or two, for example to describe the substitution function in Section 4.1, or to more precisely describe the form of weak supervision that the authors employ in Section 4.2.

As a final note, the metareviewer would suggest more explicit discussion on what authors think a VAE *should* do when factors are correlated. Arguably learning factors that reflect correlations is the "correct" when the training data exhibits such correlations. Currently the authors do not provide much of an arguments for *why* they think a VAE should learn a representation that ignores these correlations. A possible argument is that train-time correlations might not be representative of test-time correlations. Here, testing to what extent a learned representation generalizes to test-time data with a shift in correlation would also strengthen results.

On balance, the metareviewer's assessment is that this paper falls narrowly below the threshold for acceptance. While the experimental evaluation represents a substantial effort that in itself is above the bar, there are problems with narrative and the clarity of  writing that rise above the level of minor revisions that could be addressed by camera ready without additional review. Based on this, the metareviewer will recommend rejection. With a little bit more work on writing and exposition, this will make a great paper at a future conference.